# On the Measurement of Laser Lines in 3D Space with Uncertainty Estimation

**DOI:** 10.3390/s25020298

**Published:** 2025-01-07

**Authors:** Ivan De Boi, Nasser Ghaderi, Steve Vanlanduit, Rudi Penne

**Affiliations:** InViLab, Department of Electromechanical Engineering, University of Antwerp, Groenenborgerlaan 171, 2020 Antwerp, Belgium; nasser.ghaderi@uantwerpen.be (N.G.); steve.vanlanduit@uantwerpen.be (S.V.); rudi.penne@uantwerpen.be (R.P.)

**Keywords:** laser line measurement, uncertainty propagation, uncertainty quantification, Gaussian processes, calibration, Galvanometric laser scanner, Gaussian mixture model, lines in 3D space

## Abstract

Laser-based systems, essential in diverse applications, demand accurate geometric calibration to ensure precise performance. The calibration process of the system requires establishing a reliable relationship between input parameters and the corresponding 3D description of the outgoing laser beams. The quality of the calibration depends on the quality of the dataset of measured laser lines. To address this challenge, we present a stochastic method for measuring the coordinates of these lines, considering both the camera calibration uncertainties and measurement noise inherent in laser dot detection on a detection board. Our approach to composing an accurate dataset of lines utilises a standard webcam and a checkerboard, avoiding the need for specialised hardware. By modelling the uncertainties involved, we provide a probabilistic description of the fitted laser line, enabling quality assessment of the measurement and integration into subsequent algorithms. We also offer insights into the optimal number of board positions and the number of repeated laser dot measurements, which are both the main time-consuming factors in practice. In summary, our proposed method represents a significant advancement in the field of laser-based system calibration, offering a robust and efficient solution.

## 1. Introduction

Laser-based systems are used in a myriad of application domains such as 3D scanning [1], radar [2], LiDAR mechanisms [3,4], laser Doppler vibrometry [5], optical tweezers [6], medical imaging [7] and additive manufacturing [8]. Moreover, the usage of straight lines, not necessarily from laser beams, has been well-studied in the field of camera calibration [9,10,11].

The calibration of these laser-based systems means determining the relationship between controllable inputs and resulting straight lines representing the laser beams. For example, in a galvanometric laser system, a laser beam is guided by two rotating mirrors, as depicted in Figure 1. Calibrating this device means learning the relationship between the two rotation angles of the mirrors and straight lines in 3D space themselves. This mapping allows us to infer a new straight line that corresponds to a given pair of rotation angles. This calibration challenge is investigated in the works [12,13,14]. All of these rely on a dataset of known pairs of angles and measured straight lines. A similar reasoning can be given for other laser-based devices. We are only interested in a geometric description of a straight line: a point and a direction.

Even though an accurate dataset of lines is of utmost importance for the calibration of laser-based systems, a search of the literature revealed few studies which address the topic of measuring straight lines in 3D space. The reader is not to confuse this with measuring straight lines in an image, e.g., the contours of a building, which is a well-studied topic in computer vision [15,16,17].

The main challenge is that straight lines, or laser beams, cannot be captured as a whole directly. We are forced to rely on the observable intersection of the laser beam with a surface. We will follow the approach in [12,13,14], where a fixed laser line intersects a series of checkerboards, whose position is known via a camera calibration procedure. This approach is also taken in [18], where both the camera and the laser are controlled by a galvanometric setup to allow for a much wider range sensing. Alternatively, the authors in [19] propose a method to calibrate the laser beam direction based on the reflections measured on a metallic sphere. However, we are interested in more than just the direction of the laser beam. We also need to determine its position in 3D space. Another approach, which does not depend on camera calibration, is given in [20]. The authors propose a method that utilises both a laser tracker and a theodolite to ascertain the 3D coordinate of the intersection of a laser beam with a target plane. Our method does not depend on this extra hardware. Finally, a holistic calibration approach involves optimising the entire laser scanning system by determining an optimal set of parameters for its underlying mechanical model. An example of this is presented in [21], where the authors estimate and refine device parameters by minimising the normal distances between corresponding planar patches extracted from multiple views. However, our focus lies on the laser lines themselves, as they are the fundamental output of these devices. This approach aligns more closely with the physical reality, where observed points are incidental by-products of the underlying line generation process. The calibration techniques demonstrated in [12,13,14] are specifically designed to leverage line-based information for calibration purposes.

This work focuses on a method to obtain a description for a line given by a reference point *p* and a direction *d*, both stochastic in nature. The result of our method is a combination of these two probability distributions. Both are a trivariate normal (Gaussian) distribution, given by a 3×1 mean vector μ and a 3×3 covariance matrix Σ, which represents the uncertainty. Further details are given in Section 2. Having an uncertainty measurement for an inferred straight line has two major benefits. First, it allows for a quality assessment of the predicted straight line. This plays a vital role in situations where risk assessment is paramount, such as medicine. Second, this uncertainty can be exploited in subsequent algorithms that rely on these uncertainties, such as Bayesian Optimisation [22] or provide a statistically sound answer as to whether or not a point lies on a line or even if two lines intersect. We define two sources of uncertainty when measuring laser beams: the camera and the way we observe the laser beam intersecting detection boards.

The first source of uncertainty is caused by the camera itself. Not only is the image distorted due to both lens and perspective distortions, but the estimated intrinsic and extrinsic camera parameters are subject to measurement noise themselves. These are obtained through a camera calibration procedure, which is based on images. These images are noisy measurements. Even a perfect camera suffers from quantisation noise due to pixelation. To obtain a quantisation of the uncertainty of an image, we adopt the work in [10]. The authors propose to perform a pre-processing step to remove all distortions by mapping a real image of a checkerboard to a virtual perfect checkerboard. Afterwards, the camera is calibrated using a simplified version of Zhang’s method [23]. This mapping is learned by training a Gaussian process (GP) on the correspondences between checkerboard corners in a real image and the corners of a virtual checkerboard in a perfect grid. For more details, we refer to the paper itself. The main benefit of this method is the usage of a Gaussian process. A GP is a probabilistic and non-linear machine-learning regression technique [22]. The resulting predictions consist of both a mean and a variance. In other words, these predictions are accompanied by an uncertainty measurement, which we will exploit. This variance is one of the sources of uncertainty we will propagate to the resulting uncertainty on the fitted straight lines. It is an inherent property of the camera calibration method we implement.

Throughout this manuscript, we refer to the intersection of a laser beam and a checkerboard as a *laser dot*. When dots are observed on boards at multiple different locations, we can fit a straight line through them. This shifts the challenge to obtaining the 3D coordinates of the dots, which can be obtained by the camera calibration when working with detection boards. In Figure 2, two examples of measured laser dots can be seen. These measurement images suffer from quantisation noise, reflections, irregularities in the shape and sometimes flares. Moreover, some devices, such as the Polytec PSV400, have a pulsating laser beam, resulting in constantly changing dot shapes. These factors introduce variance in the laser dot measurements, which is the second source of uncertainty we propagate to the straight line fits.

The main contributions of this work are as follows:We propose a stochastic model to measure a reference point and a direction for a set of laser lines, based on the noisy measurements of individual dots on detection boards.These measurements are taken by a camera which is calibrated in a stochastic way, allowing for uncertainty propagation of those measurements.Our method relies only on the usage of a webcam and a checkerboard. We do not require additional hardware such as the methods in [19,20].We provide insights and recommendations on the number of board positions and the number of individual dot measurements, which are both the main time-consuming factors in practice.

The remainder of this paper is organised as follows: In Section 2, materials and methods are introduced. In Section 3, we provide the results of our experiments. Section 4 discusses the results of this study and points out the benefits and limitations of the proposed method. Finally, Section 5 concludes the article.

## 2. Materials and Methods

This section explains how the two primary sources of uncertainty in our measurements are propagated to the uncertain geometric description of a straight line. We take the following steps:The camera with which we take images of a laser dot on a detection board needs to be calibrated to correctly interpret the images. We propose a **probabilistic camera calibration** method which provides an uncertainty estimate by means of posterior variance of the predictions.When we repeatedly measure the location of the same laser dot in our image in pixel coordinates, we observe a spread. We are interested in the **aggregate of these measurements**, of which each individual measurement is also uncertain.The camera calibration allows us to calculate the **3D coordinate of every measured laser spot**, as the dots are located on the boards used to calibrate the camera. We also propagate the combined uncertainty to the 3D coordinates themselves.Finally, we perform a **best line fit on these 3D points**, weighted by their uncertainty.

### 2.1. Camera Calibration

In this work, we build on the camera calibration method proposed by De Boi et al. in [10]. This method first removes camera distortion and then performs a simplified version of Zhang’s method [23] on the undistorted images of a series of rotated and translated checkerboards. The distortion is captured by a Gaussian process [22], which in this context is used as a probabilistic regression technique.

In regression, we are interested in the function between given inputs and measured outputs. Once this function has been learned, we can infer new values for new unseen inputs. A Gaussian process is a probability distribution over functions, extending the concept of multivariate Gaussian distributions to infinite dimensions. It represents a distribution over possible functions that fit a set of data points. A Gaussian process prediction consists of a mean and a variance. The latter can be thought of as an uncertainty estimate accompanying the mean. In this work, we will build on that uncertainty estimate and propagate it through the methods to the description of the measured laser line.

In the first step of the camera calibration procedure, the corners of a checkerboard are detected in only one image. Next, a Gaussian process is trained between the pixel uv-coordinates of these corners and the xy-coordinates on a virtual ideal perfect squared checkerboard. For an in-depth treatise of GPs, we refer the reader to both [10] and the book [22]. We use integer values for the xy-coordinates of the checkerboard corners. Once trained, the GP maps the distortion of the camera from the real image to the virtual perfect checkerboard. By pushing any image through this GP, we obtain new xy-locations for each pixel, effectively removing all distortions. We end up with a virtual GP camera that acts like a perfect pinhole model. When conducting this for a number of images of a rotated and translated checkerboard, we can calibrate this virtual camera. An example of an undistorted image is given in Figure 3.

As mentioned above, we perform the calibration of this virtual perfect pinhole camera by Zhang’s method [23]. The camera matrix composed of the intrinsic parameters of a pinhole camera K can be written as
(1)$$K=\left(\begin{array}{ccc} f & 0 & u_{c}\\ 0 & f & v_{c}\\ 0 & 0 & 1 \end{array}\right),$$
in which *f* is the focal length, and $\left(u_{c},v_{c}\right)$ is the coordinate of the principle point. For more details, the reader is referred to [23].

In Zhang’s method, we construct a coordinate system for every checkerboard where the *Z*-axis is perpendicular to it, effectively making every *z*-component zero. This allows us to write the projection of a homogeneous world coordinate (X,Y,Z,1) to the homogeneous pixel coordinate (x,y,1) in the undistorted image
(2)$$\left(\begin{array}{c} x\\ y\\ 1 \end{array}\right)∼K\left[R|t\right]\left(\begin{array}{c} X\\ Y\\ 0\\ 1 \end{array}\right)=K\left[r_{1}|r_{2}|t\right]\left(\begin{array}{c} X\\ Y\\ 1 \end{array}\right),$$
in which $r_{1}$ and $r_{2}$ are the first two columns of R. This equation shows a 2D to 2D correspondence known as a homography H, given by
(3)$$\left(\begin{array}{c} x\\ y\\ 1 \end{array}\right)∼H\left(\begin{array}{c} X\\ Y\\ 1 \end{array}\right).$$
This matrix is only determined up to a scalar factor. From correspondences between the corner locations in the undistorted images of the checkerboards and the virtual ideal checkerboard itself, we can solve for an H. These homographies, one for every position of the checkerboard, can be decomposed in the camera intrinsics and extrinsic parameters by exploiting the fact that R is orthonormal.

The reference frame of the virtual GP-cam can be easily related to any real 3D world reference frame by a few correspondences. The scale depends on the unit chosen for the ideal checkerboard. By choosing a unit that equals the distance between two subsequent corners on the checkerboard, we enforce the virtual world to have the same scale as the real world. The principle axis, which is the *Z*-axis, is always perpendicular to this board.

The main benefit of using a Gaussian process to remove the distortion is that we can exploit the posterior variance of the prediction of the new pixel location. This can be interpreted as an uncertainty measurement for the resulting pixel coordinates of the detected corners. This uncertainty enables us to calculate an uncertain homography H between the ideal checkerboard and each of the the undistorted images of the checkerboards.

Therefore, we implement the method *Homography from Pairs of Uncertain and Fixed Points* described on page 426 in the book [24]. The detected uv-coordinates of the corners in a checkerboard are mapped to xy-coordinates by the Gaussian process model. In practice, we implement two independent GPs, one for uv to *x* and one for uv to *y*. Each GP prediction consists of a mean value μ and a covariance $\sigma^{2}$. For an image of checkerboard *n*, we write the coordinates of the *i*th detected corner as
(4)$$c_{n,i}=\left\{\begin{array}{c} \mu_{n,i},\Sigma_{n,i} \end{array}\right\}=\left\{\begin{array}{c} \left[\begin{array}{c} \mu_{x}\\ \mu_{y} \end{array}\right],\left[\begin{array}{cc} {\sigma_{x}}^{2} & 0\\ 0 & {\sigma_{y}}^{2} \end{array}\right] \end{array}\right\}.$$
The off-diagonal zeros in the covariance matrix indicate the independence of the prediction for the *x*- and *y*-coordinate by the GPs. The coordinates of the corners on the virtual ideal checkerboard are integer values with zero uncertainty. We denote for the *i*th corner on the ideal board
(5)$$c_{id,i}=\left\{\begin{array}{c} \mu_{id,i},\Sigma_{id,i} \end{array}\right\}=\left\{\begin{array}{c} \left[\begin{array}{c} \mu_{x}\\ \mu_{y} \end{array}\right],\left[\begin{array}{cc} 0 & 0\\ 0 & 0 \end{array}\right] \end{array}\right\}.$$

The following relations hold for every board *n*:(6)$$c_{n,i}=H_{id\to n}c_{id,i},$$
(7)$$c_{id,i}=H_{n\to id}c_{n,i},$$
where $H_{id\to n}$ and $H_{n\to id}$ are stochastic in nature and thus given by both a 3×3 mean matrix and a 9×9 covariance matrix. We use the notation
(8)$$H=\left\{\begin{array}{c} \mu_{H},\Sigma_{\mathrm{hh}} \end{array}\right\},$$
in which h is a 9×1 vector composed of the columns of the 3×3 matrix h.

For the camera calibration, we need only $H_{id\to n}$. However, we are also interested in the inverse of this matrix $H_{id\to n}^{-1}$. This is necessary to calculate the 3D coordinates of laser dots in Section 2.3. Inverting a stochastic matrix, including the covariance, is not trivial. The inverse relationship is non-linear. However, in this context, we can calculate it directly. We introduce the following approximation
(9)$$H_{id\to n}^{-1}∼H_{n\to id},$$
which is true for homographies without deviations on the corresponding points and without uncertainty.

We can solve for H in Equations (Equation 6) and (Equation 7) by turning $c_{n,i}$ and $c_{id,i}$ into spherically normalised homogeneous coordinates and using a non-linear Gauss–Markov model. The details of this are beyond the scope of this work and are further explained in the book [24].

Finally, for each of the *n* boards, we can decompose $H_{id\to n}^{-1}$ into intrinsic and extrinsic parameters in Equation (Equation 2). These values are needed to calculate the 3D locations of the laser dots in Section 2.3. To maintain the stochastic nature of each of these parameters, we first take a significant number of samples from every stochastic matrix $H_{id\to n}^{-1}$. Then, we apply Zhang’s method to each set of samples, one for each of the *n* boards, to obtain a value for K, $R_{n}$ and $t_{n}$. Afterwards, we calculate a mean and covariance for each of these thousand values to obtain a stochastic version of K, $R_{n}$ and $t_{n}$. This implies we have both a mean and a covariance matrix for these parameters. The latter will be used to propagate uncertainty to the calculated 3D coordinates of the laser dots in Section 2.3.

### 2.2. Measuring Dots on Boards

To estimate the uncertainty of a measured laser dot, we observe it multiple times and analyse the spread of the measurements.

In practice, we make a video of a laser being guided by the two mirrors, as seen in Figure 1. The mirrors are held stationary for a fixed time at a specific angle pairs. This provides us with an image per frame of the video of each laser dot location, resulting in multiple images of the same laser dot location. We make one film per board location. For this paper, we use MATLAB R2023b. The 2D uv-coordinate of the laser dot in every frame is determined by the built-in *util_findlaser* function of Mathworks’ Image Acquisition Toolbox. We filter out readings that are clearly wrong, such as values outside a certain region, all zero values, values with few neighbouring points, etc.

The function *util_findlaser* analyses the input image frame to identify and locate the laser point. It returns the x- and y-pixel coordinates corresponding to the laser point’s centroid. If no laser point is detected, NaN values are returned instead. Additionally, it generates a binary matrix indicating potential laser locations. The algorithm employs a four-step process:It examines the red (or green) channel of the RGB image to find the highest intensity value present.A binary image is created by identifying all locations where this maximum value occurs, representing potential laser positions.Blob analysis is performed on this binary image to identify the largest connected component. The centroid of this component is considered the laser’s location, expressed in pixel coordinates.A final verification step ensures that the pixel count in the identified blob exceeds a certain threshold. This helps eliminate false positives caused by random noise in the image.This method allows for robust laser point detection while minimising the impact of image noise and other potential sources of error.

Next, we apply k-means clustering to group measurements that belong to the same laser dot. In Figure 4, we depict the result of this filtering and k-means clustering pre-processing step for a single board that registered 9×9 laser dots.

After this, we end up with a cluster of *j* measured uv-coordinates per laser dot. In what follows, we calculate a mean and a covariance for every cluster that corresponds to a laser dot we are interested in. An example of such a cluster can be seen in Figure 5.

These uv-coordinates of pixels are now mapped to the virtual xy-world by the Gaussian process model. This mapping was learned in the first step when calibrating the GP camera and removing all distortion. The resulting xy-coordinates of the *j*th measurement for the *i*th dot on board *n* are accompanied by an uncertainty estimate in the form of a covariance matrix:(10)$$d_{n,i,j}=\left\{\begin{array}{c} \mu_{n,i,j},\Sigma_{n,i,j} \end{array}\right\}=\left\{\begin{array}{c} \left[\begin{array}{c} \mu_{x}\\ \mu_{y} \end{array}\right],\left[\begin{array}{cc} {\sigma_{x}}^{2} & 0\\ 0 & {\sigma_{y}}^{2} \end{array}\right] \end{array}\right\}.$$
The Gaussian process model predicts both this mean $\mu_{n,i,j}$ and this covariance matrix $\Sigma_{n,i,j}$. A visualisation of this is given in Figure 3c, where not only the uv-coordinates of the measured dots but in fact every pixel in the undistorted image is mapped to a new location in the virtual xy-space which is $\mu_{n,i,j}$ and also gives an uncertainty estimation which is the mean of the two GP posterior standard deviations $\sigma_{x}$ and $\sigma_{y}$ normalised to one.

This means we have a 2D Gaussian distribution for every measurement in the cluster belonging to a laser dot. In Figure 5, the red stars are at location $\mu_{n,i,j}$ and the Gaussian process posterior uncertainty is depicted by the green ellipses. The overall cluster can thus be seen as a 2D Gaussian mixture model, with as many 2D Gaussian distributions as there are video frames capturing the dot. The aggregate of each cluster can be represented by a 2D mean and a 2×2 covariance matrix.

The resulting mean of the cluster can naively be calculated as just the mean of the individual measurements. However, as we have an uncertainty estimate of every measurement, we can nuance this approach by taking a weighted mean. We use the Gaussian process posterior uncertainties, i.e., the variances, to calculate the weight ${\hat{w}}_{n,i,j}$ for a single measured dot $d_{n,i,j}$ as follows:(11)$${\hat{w}}_{n,i,j}=\frac{1}{\sqrt{\sigma_{x}^{2}+\sigma_{y}^{2}}}.$$
The square root of the trace of the covariance matrix (the denominator in the equation above) is known as the Helmert point error [24]. It characterises the uncertainty by a single number. Next, we divide this weight by the sum of all weights, so that the resulting weights add up to one.
(12)$$w_{n,i,j}=\frac{{\hat{w}}_{n,i,j}}{\sum_{j}{\hat{w}}_{n,i,j}}.$$
Using these weights we calculate the weighted mean of the measured dots for the *i*th cluster on board *n*
(13)$$\mu_{n,i}=\sum_{j}w_{n,i,j}\mu_{n,i,j}.$$
The cluster variance is not simply the variance of the means, as this would ignore the GP variance of each individual dot measurement. We calculate the overall covariance matrix for the *i*th cluster on board *n* as
(14)$$\Sigma_{n,i}=\sum_{j}w_{n,i,j}\left(\Sigma_{n,i,j}+\mu_{n,i,j}\mu_{n,i,j}^{T}\right)-\mu_{n,i}\mu_{n,i}^{T}.$$

Thus, for a cluster of measurements belonging to the *i*th laser dot on board *n*, which we consider a Gaussian mixture model, we obtain as aggregate for the xy-coordinates
(15)$$d_{n,i}=\left\{\begin{array}{c} \mu_{n,i},\Sigma_{n,i} \end{array}\right\}=\left\{\begin{array}{c} \left[\begin{array}{c} \mu_{x}\\ \mu_{y} \end{array}\right],\left[\begin{array}{cc} {\sigma_{x}}^{2} & \rho \sigma_{xy}\sigma_{y}\\ \rho \sigma_{xy}\sigma_{y} & {\sigma_{y}}^{2} \end{array}\right] \end{array}\right\},$$
in which ρ is the correlation between the *x*- and *y*-coordinate. We consider this aggregate the 2D measurement of a laser dot on a board with uncertainty. An example is given in Figure 5. The red stars are the means of the Gaussian process predictions of every individually measured dot, which is one per frame of the video capturing the dot. The green ellipses have principle axes whose sizes correspond to the Gaussian process variance reformulated as two times sigma. The red ellipse is fitted on the spread of the means (red stars), also depicting the two times sigma region. The blue ellipse is fitted on the same spread of the means, but taking the Gaussian process uncertainty (the green ellipses) into account using Equation (Equation 11). There is more uncertainty when taking the green circles into account. Also, the orientation between the blue ellipse and the red ellipse is slightly different.

### 2.3. 3D Coordinates of Dots

Once we have a calibrated camera, and thus know where the boards are in 3D, and we have a 2D pixel coordinate of a laser dot, we can calculate its 3D coordinates. Moreover, both the camera calibration and the 2D measurement have an uncertainty representation in the form of a covariance matrix, which we can propagate to the 3D world. The end result will be a 3D coordinate of a laser dot with an uncertainty estimate. Again, we follow the reasoning in [24].

In Section 2.2 we denoted the stochastic 2D non-homogeneous coordinate of the *i*th measured laser dot on board *n* with the Gaussian process model undistorted image as $d_{n,i}$. To simplify the notation below, we will refer to it as *x*. In fact, the following holds for any generic point in the virtual GP-camera image, not just the measured laser dots. We can write
(16)$$x=\left\{\begin{array}{c} \mu_{x},\Sigma_{xx} \end{array}\right\}=\left\{\begin{array}{c} \left[\begin{array}{c} \mu_{x}\\ \mu_{y} \end{array}\right],\left[\begin{array}{cc} {\sigma_{x}}^{2} & \rho \sigma_{xy}\sigma_{y}\\ \rho \sigma_{xy}\sigma_{y} & {\sigma_{y}}^{2} \end{array}\right] \end{array}\right\}.$$

We make the 2D coordinate *x* homogeneous and normalise them to Euclidean form to create coordinate x by writing
(17)$$x=\left\{\begin{array}{c} \mu_{x},\Sigma_{xx} \end{array}\right\}=\left\{\begin{array}{c} \left[\begin{array}{c} \mu_{x}\\ 1 \end{array}\right],\left[\begin{array}{cc} \Sigma_{xx} & 0\\ 0^{T} & 0 \end{array}\right] \end{array}\right\}.$$
In essence, we just added a 1 to the mean vector and a row and column of zeros to the covariance matrix $\Sigma_{xx}$. Note how this covariance matrix is no longer positive definite. There is no uncertainty about the third component of the mean vector, which is always exactly 1.

This homogeneous 2D coordinate x can now be mapped to the ideal virtual board where the Gaussian process was trained in Section 2.1 via uncertain homography $H_{n\to id}$. We obtain a point $x^{\prime }$ given by
(18)$$x^{\prime }=\left\{\begin{array}{c} \mu_{x^{\prime }},\Sigma_{x^{\prime }x^{\prime }} \end{array}\right\}=\left\{\begin{array}{c} \mu_{H_{n\to id}}x\;,\;\mu_{H_{n\to id}}\Sigma_{xx}{\left(\mu_{H_{n\to id}}\right)}^{T}+\left(x^{T}⊗I_{3}\right)\Sigma_{\mathrm{hh}}\left(x⊗I_{3}\right) \end{array}\right\}.$$
Notice how the resulting uncertainty of $x^{\prime }$ is given by two terms. The first is the result of the uncertainty in the measurement of the dots $\Sigma_{xx}$. The second is due to the uncertainty of the camera calibration $\Sigma_{\mathrm{hh}}$. The point with coordinates $x^{\prime }$ can now be brought to each of the checkerboards in 3D, as the camera calibration in Section 2.1 also yields a $R_{n}$ and $t_{n}$ for each of the *n* boards.

Next, we normalise $x^{\prime }$ in the Euclidean sense again so that the homogeneous component of the coordinate’s mean vector is 1. We can write any homogeneous vector x as the combination of a Euclidean part $x_{0}$ and a scalar homogeneous part $x_{h}$ as follows
(19)$$x=\left[\begin{array}{c} x_{0}\\ x_{h} \end{array}\right].$$
We follow the notation in book [24], where the homogeneous part is the last component of the homogeneous vector. Considering the homogeneous part of $\mu_{x^{\prime }}$ as $\mu_{h}$, we can write
(20)$$x^{\prime }∼\left\{\begin{array}{c} \frac{\mu_{x^{\prime }}}{\mu_{h}},\frac{1}{{\left(\mu_{h}\right)}^{2}} \end{array}\right\}\Sigma_{x^{\prime }x^{\prime }}.$$

To obtain the coordinates of a point in 3D, we first set the homogeneous component to zero. This operation can be seen as
(21)$$x_{0}^{\prime }=\left\{\begin{array}{c} \mu_{x_{0}^{\prime }},\Sigma_{x_{0}^{\prime }x_{0}^{\prime }} \end{array}\right\}=\left\{\begin{array}{c} A\mu_{x^{\prime }}\;,\;A\Sigma_{x^{\prime }x^{\prime }}A^{T} \end{array}\right\},\;\mathrm{with}\;A=\left[\begin{array}{ccc} 1 & 0 & 0\\ 0 & 1 & 0\\ 0 & 0 & 0 \end{array}\right]$$
We apply $R_{n}$ and $t_{n}$ to this homogeneous coordinate in a single matrix multiplication with a 4×4 matrix $H_{n}$. This is not to be confused with matrices $H_{id\to n}$ and $H_{n\to id}$, who are 3×3. The matrix $H_{n}$ maps points in 3D to other points in 3D while preserving co-linearity. This means it is a homography in 3D, hence the usage of the letter *H*. As this only applies to homogeneous coordinates, we first rewrite $x_{0}^{\prime }$ as a Euclidean normalised homogeneous 3D coordinate
(22)$$x_{0}^{e}=\left\{\begin{array}{c} \mu_{x_{0}^{e}},\Sigma_{x_{0}^{e}x_{0}^{e}} \end{array}\right\}=\left\{\begin{array}{c} \left[\begin{array}{c} \mu_{x_{0}^{\prime }}\\ 1 \end{array}\right],\left[\begin{array}{cc} \Sigma_{x_{0}^{\prime }x_{0}^{\prime }} & 0\\ 0^{T} & 0 \end{array}\right] \end{array}\right\}.$$
The 4×4 covariance matrix $\Sigma_{x_{0}^{e}x_{0}^{e}}$ is not positive definite and the zeros are the result of the absence of uncertainty on the homogeneous part of $\mu_{x_{0}^{e}}$, which is exactly 1.

We construct the matrix $H_{n}$ out of $R_{n}$ and $t_{n}$, which are all stochastic variables resulting from the approach of the camera calibration in Section 2.1. For every board *n*, we have
(23)$$H_{n}=\left\{\begin{array}{c} \mu_{H_{n}},\Sigma_{\mathrm{hh}} \end{array}\right\},$$
We construct the mean matrix $\mu_{H_{n}}$ from the individual mean values of the 3×3 $\mu_{R_{n}}$ and the 3×1 $\mu_{t_{n}}$ as follows
(24)$$\mu_{H_{n}}=\left[\begin{array}{cc} \mu_{R_{n}} & \mu_{t_{n}}\\ 0^{T} & 1 \end{array}\right].$$
The 16×16 covariance matrix $\Sigma_{\mathrm{hh}}$ is calculated as the covariance of all *j* samples off $R_{n}$ and $t_{n}$, and rewritten using eq:MuH into *j* 4×4 matrices. Each one of these matrices is then rewritten as 1×16 row vector $h^{T}$. We obtain
(25)$$\Sigma_{\mathrm{hh}}=\mathrm{Cov}\left(h_{1}^{T};...;h_{j}^{T}\right).$$
Next, we can calculate the 3D homogeneous coordinate X as
(26)$$X=\left\{\begin{array}{c} \mu_{X},\Sigma_{XX} \end{array}\right\}=\left\{\begin{array}{c} \mu_{H_{n}}x_{0}^{e}\;,\;\mu_{H_{n}}\Sigma_{x_{0}^{e}x_{0}^{e}}{\left(\mu_{H_{n}}\right)}^{T}+\left({\left(x_{0}^{e}\right)}^{T}⊗I_{4}\right)\Sigma_{\mathrm{hh}}\left(x_{0}^{e}⊗I_{4}\right) \end{array}\right\}.$$
Again, the resulting uncertainty $\Sigma_{XX}$ is given by two terms. The first is the result of the uncertainty of the dot’s location. The second is due to the uncertainty of the camera calibration. We perform an additional scaling to this resulting coordinate to have its homogeneous component of the mean vector at a value of 1. We denote the homogeneous part of $\mu_{X}$ as $\mu_{h}$ and write
(27)$$X∼\left\{\begin{array}{c} \frac{\mu_{X}}{\mu_{h}},\frac{1}{{\left(\mu_{h}\right)}^{2}}\Sigma_{XX} \end{array}\right\}.$$

We are interested in the non-homogeneous 3D coordinate *X*. For the mean, we can retain the Euclidean component $\mu_{X_{0}}$ of the homogeneous vector $\mu_{X}$ by dropping the homogeneous component $\mu_{h}$. For the uncertainty, we also consider the Jacobian. This yields
(28)$$X=\left\{\begin{array}{c} \mu_{X},\Sigma_{XX} \end{array}\right\}=\left\{\begin{array}{c} \mu_{X_{0}}\;,\;J^{T}\left(\mu_{X}\right)\Sigma_{XX}J\left(\mu_{X}\right) \end{array}\right\},$$
with the transpose of Jacobian J(x)
(29)$$J^{T}\left(x\right)=\frac{1}{x_{h}^{2}}\left[\begin{array}{c} x_{h}I_{3}|-x_{0} \end{array}\right],$$
evaluated at the mean $\mu_{X}$.

At the end of these steps, we have a 3D coordinate $X_{n,i}$ for the *i*th laser dot observed on board *n*. Moreover, this stochastic coordinate is accompanied by an uncertainty estimate, which we exploit in the next section when we perform the best line fit through these points.

### 2.4. Uncertain Line Through Uncertain Points

We aim for a stochastic representation of a line *L*. We group all points with coordinates $X_{n,i}$ by the pair of mirror rotations that produced that line *L*. As demonstrated in Figure 1, a pair of rotations yield a single laser beam, which is the straight line we are interested in. For the *l*th line, we denote the *n*th point on that line as
(30)$$X_{l,n}=\left\{\begin{array}{c} \mu_{X_{l,n}},\Sigma_{X_{l,n}} \end{array}\right\}.$$
In this work, we simultaneously measure all the lines represented by a set of mirror angle pairs. We describe the *l*th line *L* by the combination of a reference point $p_{l}$ and a direction $d_{l}$, which are both stochastic in nature.

To determine the reference point $p_{l}$ for the *l*th line, we first calculate a weight for every point $X_{l,n}$ based on the reciprocal of the square root of the trace of the covariance matrix $\Sigma_{X_{l,n}}$.
(31)$${\hat{w}}_{l,n}=\frac{1}{\sqrt{\sigma_{x}^{2}+\sigma_{y}^{2}+\sigma_{z}^{2}}}.$$
As mentioned before, the denominator is also known as the Helmert point error [24]. Next, we divide this weight by the sum of all the *n* weights, so that the resulting weights add up to one.
(32)$$w_{l,n}=\frac{{\hat{w}}_{l,n}}{\sum_{n}{\hat{w}}_{l,n}}.$$
Using these weights, we calculate the weighted mean of the *n* points $X_{l,n}$
(33)$$\mu_{p_{l}}=\sum_{n}w_{l,n}\mu_{X_{l,n}}.$$

The variance of $p_{l}$ is not just the weighted sum of the variance of all $X_{l,n}$, as this would ignore the variance of the individual 3D points and the fact that these points are highly correlated. These points are not independent, as they are produced by $R_{n}$ and $t_{n}$ of the *n*th board they lay on, which are obtained through the camera calibration procedure explained in Section 2.1. They are linked via the resulting camera calibration matrix K. This means the covariance between the *n* points $X_{l,n}$ has to be taken into account. We calculate the overall covariance matrix by
(34)$$\Sigma_{p_{l}}=\sum_{i=1}^{n}w_{l,i}^{2}\Sigma_{X_{l,i}}+\sum_{i=1}^{n}\sum_{j\neq 1}w_{l,i}w_{l,j}\mathrm{Cov}\left(X_{l,i},X_{l,j}\right).$$
where the second term captures the dependence between the *n* points. This covariance between two stochastic points $X_{l,i}$ and $X_{l,j}$ has to be obtained from the data themselves, i.e., the individual measured dots after being mapped to xy-space by the Gaussian process. The overall covariance $\Sigma_{p_{l}}$ is larger than in the case of independent points.

Finally, we obtain a stochastic 3D coordinate for a reference point for the *l*th line given by
(35)$$p_{l}=\left\{\begin{array}{c} \mu_{p_{l}},\Sigma_{p_{l}} \end{array}\right\}.$$

To determine the stochastic direction $d_{l}$ of the *l*th line *L*, we aggregate all directions between the points $X_{l,n}$ on that line and the reference point $p_{l}$ for that line. For each of the points $X_{l,n}$, we calculate the mean of the normalised direction $d_{l,n}$ using
(36)$$\mu_{d_{l,n}}=\frac{\mu_{X_{l,n}}-\mu_{p_{l}}}{∥\mu_{X_{l,n}}-\mu_{p_{l}}∥}.$$
In practice, we flip the direction vector if the z-component is less than zero, so that all the direction vectors point in the direction of the Z-axis. The uncertainty of this direction is given by
(37)$$\Sigma_{d_{l,n}}=\frac{\Sigma_{X_{l,n}}+\Sigma_{p_{l}}-2\mathrm{Cov}\left(X_{l,n},p_{l}\right)}{{∥\mu_{X_{l,n}}-\mu_{p_{l}}∥}^{2}}.$$
Again, as the reference point for the line is clearly dependent on one of the points itself, we have to take the covariance between them into account. In this case, the overall uncertainty is reduced. A large correlation between a point and the reference point results in a low uncertainty for the direction obtained between the both of them.

This per direction uncertainty $\Sigma_{d_{l,n}}$ allows us to calculate weights $w_{l,n}$, by taking the Helmert point error as defined in Equation (Equation 31). Moreover, we normalise $\mu_{d_{l,n}}$ to unit length by dividing it by its norm. Following the standard rules for the normal distribution, we apply the same scaling factor squared to $\Sigma_{d_{l,n}}$. This means the weight of points farther away from the reference point is larger, as the resulting uncertainty is smaller. Points close to the reference point have an uncertainty that is scaled down less and thus has a smaller weight. The mean and covariance for the direction of the *l*th line are now given by
(38)$$\mu_{d_{l}}=\sum_{n}w_{l,n}\mu_{d_{l,n}},$$
and
(39)$$\Sigma_{d_{l}}=\sum_{n}w_{l,n}\Sigma_{d_{l,n}}.$$
Together, these form a description for the stochastic direction
(40)$$d_{l}=\left\{\begin{array}{c} \mu_{d_{l}},\Sigma_{d_{l}} \end{array}\right\}.$$

In conclusion, the combination of the reference point $p_{l}$ and the direction $d_{l}$ describe the *l*th stochastic line *L*.

## 3. Results

In this section, we experimentally evaluate the method described in Section 2. We investigate to what extent the number of measurements per dot and the number of boards affect the outcome of the found reference point $p_{l}$ and the direction $d_{l}$ for a laser line we wish to measure.

### 3.1. Datasets

To assess our method, we made two setups. The first setup measures the lines produced by a Polytec PSV400 Laser Doppler Vibrometer [25]. The resolution for the Polytec PSV400 system is 9 bits. The second setup is built around a Polytec VibroFlex Xtra. The Polytec VibroFlex Xtra is not a scanning system, so we added a galvanometric system with mirrors (model GVS012 M from THORLABS [26]) to enable the laser beam to be guided. We used an Arduino Due to control the angular positions of the two mirrors. The Arduino Due features a 12-bit resolution for each of its analogue output pins, which provides a theoretical resolution of approximately 1 millidegree within the working angular range of the galvanometer system (+1.1 degrees to +5.5 degrees). However, the resolution of the galvanometric hardware with the scanner is 15 microradians (860 microdegrees), as specified in the datasheet, which is higher than the resolution provided by the Arduino. An overview of both setups is given in Figure 6.

Using these two setups, we created four datasets of sets of lines. We provide details in Table 1. The datasets are given a name which describes the size of the grid of measured laser lines. The resulting laser lines are depicted in Figure 7. The webcam used for all datasets is an Avalue 2k Webcam. The same checkerboard depicted in Figure 3 is used throughout. It has 14×9 corners. We placed the board in ten different positions. In Figure 8 and Figure 9, the results of the camera calibration procedure are depicted for each dataset. The first column shows the placement of ten different boards. The reference frame is determined by the Gaussian process camera, which can be seen by the fact that the first board is always perfectly perpendicular to the Z-axis. The Gaussian process camera maps every pixel location from a real image to a location on a virtual image, accompanied by uncertainty. These are the mean and standard deviations of the predictions, respectively. The largest uncertainty over all pixels in the GP camera image is given in the last column of Table 1.

### 3.2. Measurements

For each of the four datasets, we varied the number of measurements per dot and the number of boards. For every combination, we calculated the stochastic 3D coordinate of the reference point $p_{l}$ and the direction $d_{l}$ for every line *L* in the datasets. Both $p_{l}$ and $d_{l}$ have a mean 3×1 vector and a 3×3 covariance matrix, as described in Equations (Equation 35) and (Equation 40), respectively.

#### 3.2.1. Uncertainty

The uncertainty of the reference point $p_{l}$ and the direction $d_{l}$ of line *L*, given by $\Sigma_{p_{l}}$ and $\Sigma_{d_{l}}$, serves as a quality assessment of the measurement. Again, we investigate the impact of the number of measurements per dot and the number of boards. As before, we summarise the uncertainty given by a covariance matrix by the Helmert point error, which is the square root of its trace. See also Equation (Equation 31). Every combination of the number of measurements per dot and the number of boards yields one Helmert point error per line for both the reference point $p_{l}$ and the direction $d_{l}$.

We plot the mean and the variance of these Helmert point errors in Figure 10. In this plot, we represent the Helmet point errors of all lines in a dataset for a given combination of the number of measurements per dot and the number of boards by a circle. The radius of the circle is determined by the mean of all these Helmert point errors. The colour of this circle depends on the variance of these Helmert point errors. We conduct this for both the reference point $p_{l}$ and the direction $d_{l}$. The numbers are divided by the maximum to normalise them between zero and one. The largest circle in a plot has a radius of one, and all other circles are scaled down relative to it. The circle belonging to the set with the highest variance is highlighted in red. A smaller circle means less uncertainty for those lines. A circle that is more blue means the uncertainties are consistent. To be able to compare the results of the different datasets in absolute numbers, we depict the boxplots of the Helmert point errors per dataset in Figure 11.

#### 3.2.2. Mean

In the absence of an absolute ground truth, we compare the resulting lines for every combination of the number of measurements per dot and the number of boards to the lines found based on the maximum available data. In our case, this is a hundred measurements per dot and ten boards. For every line in the datasets, we calculate the norm of the difference between the reference points and the difference of the directions. We take the mean and the variance of all those differences and plot them in Figure 12. Again, we normalised the radii by the maximum for that dataset. Note that the top right circle is no longer visible as it has a radius of zero. For comparison, we include the boxplots of the absolute numbers in Figure 13.

## 4. Discussion

The placement of the first board in the real world is crucial for the uncertainty of the Gaussian process. In Figure 9c for dataset 7×7 we observe that the corners of the first checkerboard, used to train the Gaussian process, do not fully cover the entire image. This results in large Gaussian process posterior uncertainties in the outer regions of the image. The pixels on the right side of the image are mapped more towards the centre. This is because the posterior prediction falls back on the zero-mean prior when the uncertainty is high. As shown in Table 1, the un-normalised maximum pixel uncertainty for the 7×7 dataset is notably higher. Conversely, datasets 9×9 and 6×6 exhibit lower maximum uncertainties due to their first checkerboards better covering the real image. The 5×5 dataset falls somewhere in between.

This large Gaussian process posterior uncertainty propagates through to the 2D pixel locations of the dots on the virtual image plane of the Gaussian process camera, and thus also to the eventual 3D locations of these dots. This eventually had an effect on the uncertainty of the reference point of the line, which is notable in the top left plot of Figure 11.

As can be seen in Figure 10, the number of measurements per dot has little effect on the Helmert point error of the reference point of a line. In the subplots of rows 1, 3, 5 and 7 the circle radii do not shrink notably with an increase in the number of measurements per dot. In contrast, the Helmert point error for the direction in rows 2, 4, 6 and 8 is much more influenced by an increase in the number of measurements per dot. An explanation for this is that the uncertainty for a reference point is largely dominated by the uncertainty over the direction of the line and to a lesser extent by the directions perpendicular to the line. Adding more measurements per dot does not change that uncertainty.

Similarly, the mean of the resulting reference point and direction also does not change significantly with an increase in the number of measurements per dot, as can be observed in Figure 12. Of course, the reference point does vary a lot with the number of boards in subplots of rows 1, 3, 5 and 7. This is mainly attributable to a shift in the z-direction when adding another board and depends highly on the placement of the boards. In rows 2, 4, 6 and 8, the trend is a decrease in the difference between the directions of the lines and the lines based on the most data.

Placing more boards leads to convergence in the descriptions of the lines, as well as a decline in uncertainty. We observe that given enough measurements per dot, after five or six boards, no significant progress is made. This also depends on the placement of the boards. In practice, one is limited in this aspect. Placing the boards further away from the origin of the laser results in points farther away, which in turn, leads to a better spread in the points themselves and thus a better base to determine the lines. However, placing boards further away also comes with two major downsides. First, the number of pixels in the image that captures the dot is smaller. This region might become too small to properly detect. Second, the laser dots have to hit the detection board. If the board is farther away, some of the lasers might no longer hit the board. This is an issue for the most extreme values of α and β, i.e., the mirror angles. On the other hand, the boards cannot be placed too close to the laser or the camera, as the entire checkerboard needs to be in the captured image.

RANSAC (Random Sample Consensus) is a very common practice to remove outliers from a set of points to fit a line to. However, we noticed that the points in our datasets were relatively collinear. Using an additional RANSAC step is not optimal for several reasons. First, RANSAC introduces unnecessary computational overhead due to its iterative and random sampling nature, which is unwarranted when dealing with predominantly collinear data. Additionally, RANSAC’s outlier rejection mechanism may inadvertently discard valid points that deviate slightly from perfect collinearity, thereby losing valuable information about the underlying linear relationship. This can lead to reduced model accuracy, as excluding these points may result in a less accurate representation of the true line. Moreover, RANSAC assumes a significant proportion of outliers, which is not the case in highly collinear datasets, violating a fundamental assumption of the algorithm. The reader is, however, advised to assess whether or not this RANSAC step is necessary.

Based on our findings across different datasets, camera resolution does not significantly impact accuracy. Instead, the calibration of the Gaussian process camera emerges as the most crucial factor. Our current implementation uses a squared exponential kernel, which is known to over-smooth non-stationary data [22]. This can be problematic, especially for images captured with fish-eye lenses, where distortion varies across the image. To address these limitations, our future work will focus on two main areas. First, we aim to investigate kernels that can better capture non-stationary distortions, particularly in the heavily warped outer regions of fish-eye lens images. This approach should improve the Gaussian process’s ability to model varying covariance across the image. Second, as noted in [10], Gaussian processes trained on checkerboard corner locations can underestimate the curvature in image peripheries due to over-smoothing. To mitigate this, we plan to explore more granular, pixel-level approaches, such as using Gray code patterns. This method should increase the number and distribution of data points, especially in the outer regions of the image, leading to more accurate calibration. These improvements aim to enhance the overall accuracy and reliability of our Gaussian process-based camera calibration method, particularly for wide-angle and fish-eye lenses.

A rigorous evaluation of our uncertainty-aware line measurement method against traditional approaches lacking uncertainty quantification is hindered by the absence of definitive ground truth. To address this limitation, we calculated the discrepancies between line estimations derived from a subset of boards and those obtained using the entire dataset. The latter, a more comprehensive set, was considered a practical reference. Notably, the Helmert point errors, as depicted in Figure 12, diminish significantly when the entire board set is used. This suggests that, in such scenarios, uncertainty becomes negligible, leading to comparable results between the uncertainty-aware and the uncertainty-agnostic approach. Figure 12 further illustrates that, with an increasing number of boards, the deviation from the reference diminishes, implying minimal differences between the two approaches. In other words, given enough boards, the uncertainty does not result in better lines. It does serve as a valuable metric for assessing measurement quality, with lower uncertainty indicating greater reliability. It answers the question of how many boards to use. Moreover, this uncertainty information can be leveraged in downstream calibration processes. For instance, in the calibration procedure of a laser Doppler vibrometer, lines characterised by higher uncertainty can be assigned reduced weight, thereby optimising the overall calibration accuracy.

## 5. Conclusions

The purpose of the current study was to determine a statistical description of a line in 3D space based on noisy measurements. This description consists of a reference point *p* and a direction *d*, both stochastic in nature. We showed how to propagate the uncertainty from the camera calibration and the measurements of the dots to these trivariate Gaussian distributions. Moreover, our method only requires a checkerboard and a webcam.

The initial board used for camera calibration significantly influences the overall uncertainty of the line description. All detected dot pixel locations should be proximate to the detected corner pixel locations. If not, a dot located far from these corners will have a Gaussian process (GP) posterior prediction with a large standard deviation. This uncertainty propagates through to the dot’s 3D location, affecting the line’s reference point and direction.

Our empirical findings reveal that increasing the number of measurements per dot significantly reduces the uncertainty, or Helmert point error, for the line’s direction, but has minimal impact on its reference point. We discovered that with sufficient measurements per dot, only a limited number of boards are necessary. Adding more boards provides little benefit, likely due to constraints imposed by camera proximity and laser source distance.

The monitoring of the uncertainty of the predicted lines decreases with the number of boards and the number of measurements per dot. This serves as an example for the evaluation of predictions when ground truth is not available. This principle can be applied in various other areas, such as LiDAR calibration or laser Doppler vibrometry, or in areas outside the use of straight lines, such as statistical shape modelling or finance.

We conclude this paper by providing some practical guidelines:Placement of calibration boards:-Ensure the first calibration board covers the entire image, especially the corners and edges.-Place subsequent boards at varying distances from the camera and laser origin.-Aim for five to six boards to achieve convergence in line descriptions and uncertainty reduction.Number of measurements:-Increasing measurements per dot significantly improves direction accuracy but has less impact on reference point accuracy.-Aim for a balance between measurement quantity and practical constraints.Board positioning: Place boards far enough from the origin to improve point spread, but consider limitations:-Ensure laser dots remain detectable in images.-Keep the entire checkerboard visible in captured images.-Avoid placing boards too far, as extreme mirror angles may cause lasers to miss the board.Higher camera resolution does not necessarily improve accuracy. Focus on proper calibration instead.For highly collinear datasets, avoid using RANSAC as it may introduce unnecessary computational overhead and potentially discard valid data points. Assess the necessity of outlier removal based on the specific dataset characteristics.Pay attention to Gaussian process posterior uncertainties, especially in image regions not well-covered by calibration boards. Be aware that high uncertainties can propagate through to 3D point locations and line reference points.

By following these guidelines, researchers can optimize their setup for more accurate and reliable results in Gaussian process-based camera calibration and laser line detection.

## Figures and Tables

**Figure 1 sensors-25-00298-f001:**
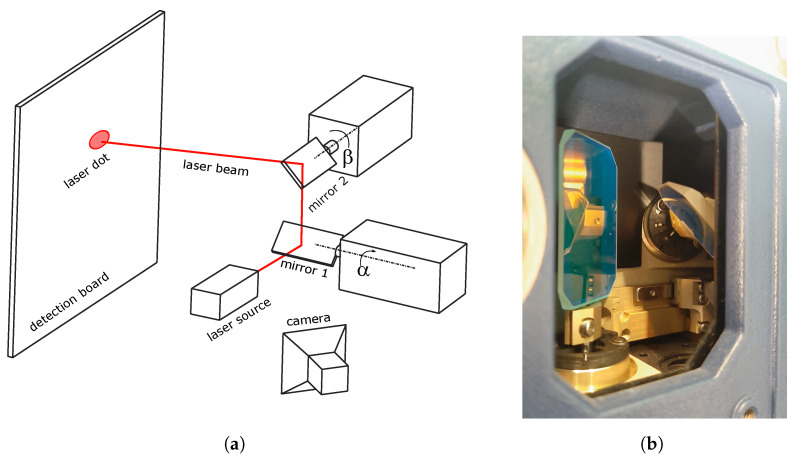
An example of a galvanometric laser setup. (**a**) A laser beam is guided by two rotating mirrors. The beam hits a detection board in a specific position resulting in a laser dot. (**b**) The two galvanic rotating mirrors in a Polytec PSV400 Scanning Vibrometer.

**Figure 2 sensors-25-00298-f002:**
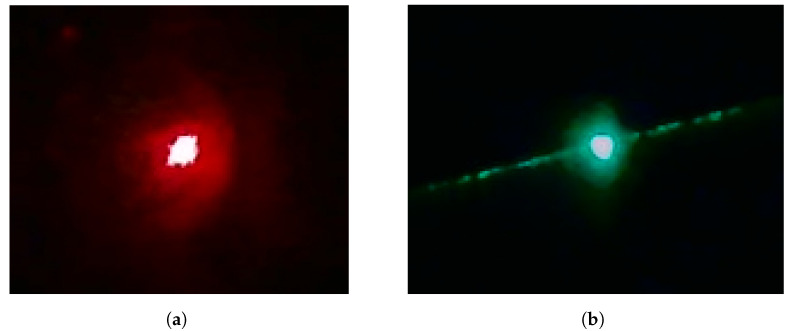
Two examples of captured laser dots. Notice the importance of the resolution, the reflections, the irregular shape and the flare for the green dot.

**Figure 3 sensors-25-00298-f003:**
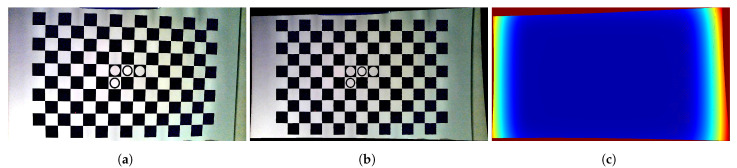
(**a**) An image of a checkerboard. Notice the slight barrel distortion. (**b**) The resulting image when undistorted and rectified by the GP model. Pixels from the original image are relocated to the mean of the GP prediction. (**c**) The standard deviation for each pixel as predicted by the GP model. Notice the low uncertainty (blue) in the vicinity of the checkerboard corners and the large uncertainty away from them (red). We normalised the values between zero and 1 by dividing every value by the maximum uncertainty.

**Figure 4 sensors-25-00298-f004:**
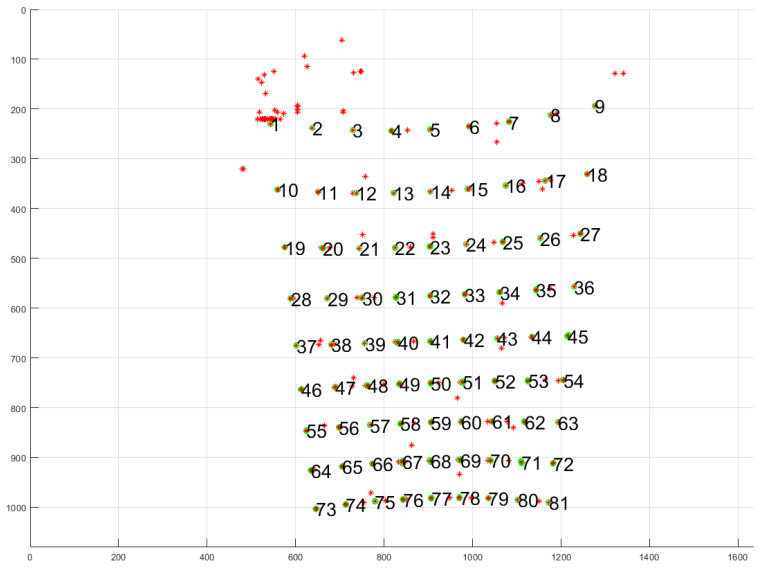
An example of the result of the filtering and k-means clustering pre-processing step for a single board that registered 9×9 laser dots. The red * are laser positions detected by *util findlaser*. The green circle around some of them means they have enough neighbours to be considered valid. The red * without this green circle are failed laser dot measurements, mostly due to lighting conditions or the fact that the laser beam moved when the galvanometer mirrors moved to another angle pair. The black numbers indicate the k-means clustering result on the red * with green circle.

**Figure 5 sensors-25-00298-f005:**
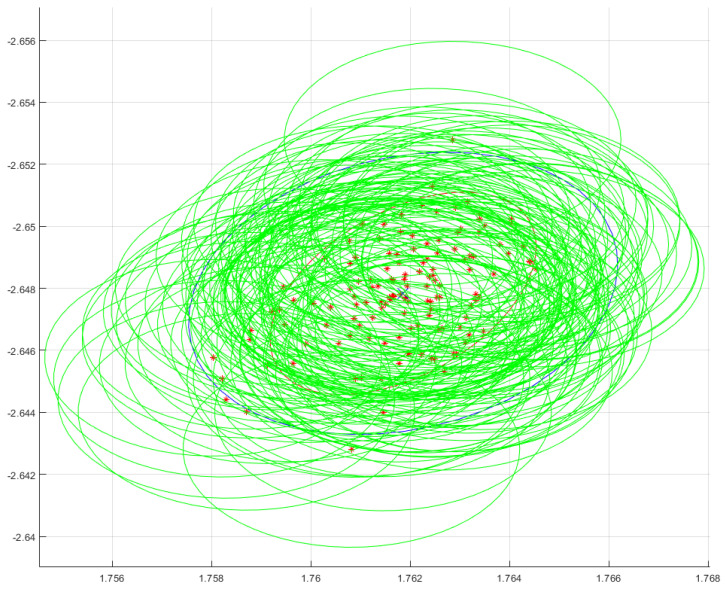
An example of a typical measurement situation of a single laser dot of interest. The red * are the means of the GP predictions. The green ellipses have principle axes whose sizes correspond to the GP variance. The red ellipse is fitted on the spread of the red stars. The blue ellipse is fitted on the same spread of the means, but taking the GP uncertainty (the green ellipses) into account using Equation (Equation 11).

**Figure 6 sensors-25-00298-f006:**
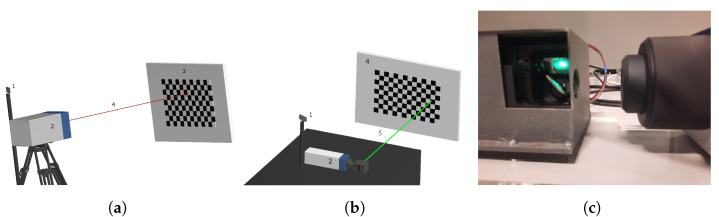
(**a**) A schematic overview of the PSV400 Scanning Vibrometer setup: 1 Webcam, 2 vibrometer, 3 checkerboard, 4 laser beam. (**b**) An overview of VibroFlex Xtra Sensor Head setup: 1 Webcam, 2 vibrometer, 3 mirrors, 4 checkerboard, 5 laser. (**c**) An image of the setup in (**b**).

**Figure 7 sensors-25-00298-f007:**
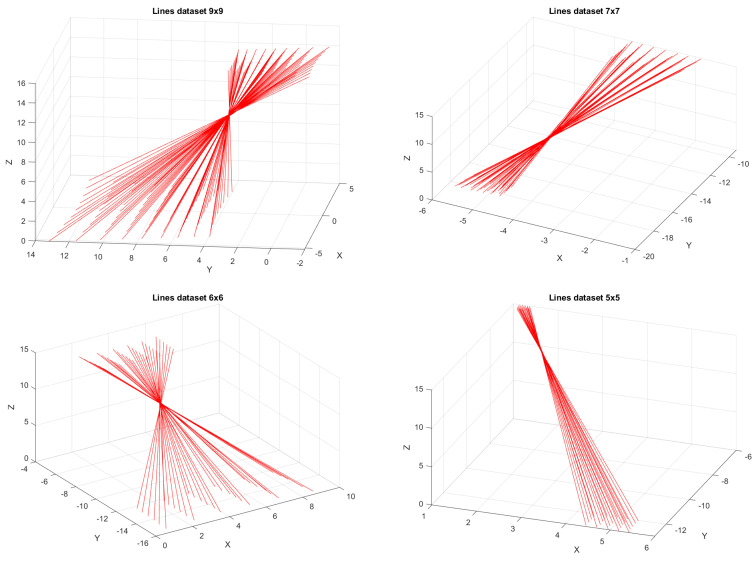
The four sets of lines in the datasets. These are plotted based on the results from using the maximum number measurements per dot and the maximum number of boards.

**Figure 8 sensors-25-00298-f008:**
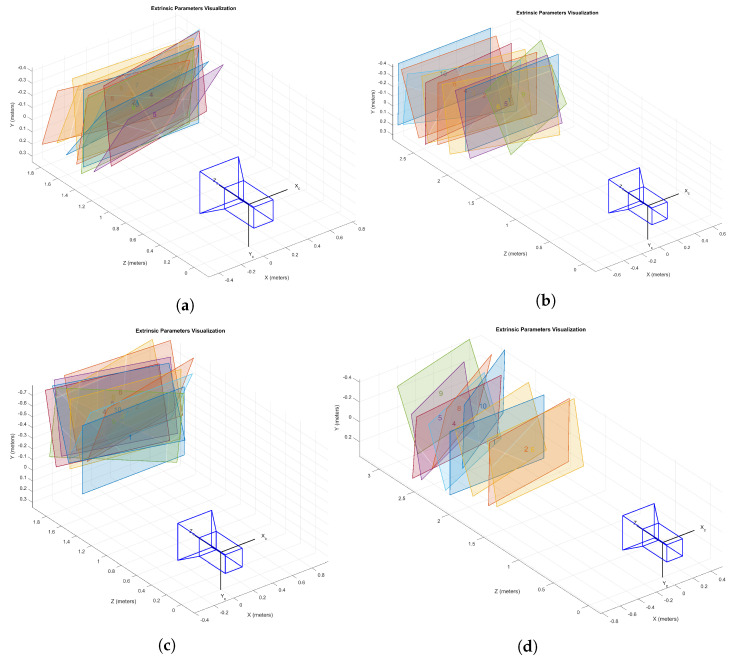
Camera calibration boards in the GP camera reference frame for datasets 9×9 (**a**), 7×7 (**b**), 6×6 (**c**) and 5×5 (**d**). The camera is depicted in the GP-camera reference frame. Each of the boards has been given a different colour. Every laser line in a dataset intersects with every board.

**Figure 9 sensors-25-00298-f009:**
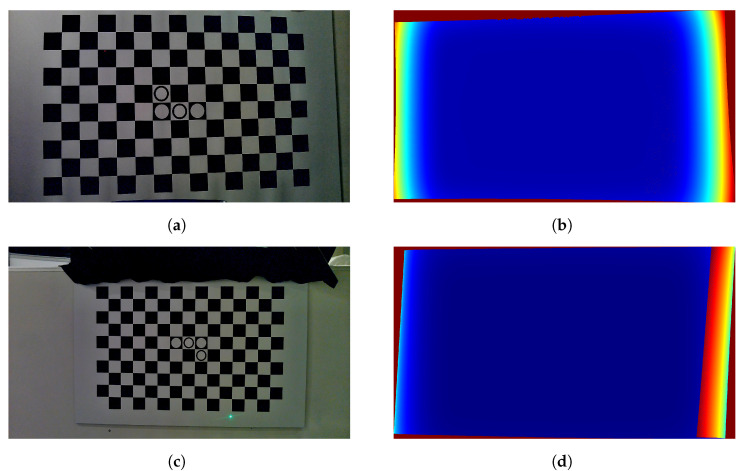
Camera calibration results for datasets 9×9 (**a**,**b**), 7×7 (**c**,**d**), 6×6 (**e**,**f**) and 5×5 (**g**,**h**). Column 1: The first board, on which the Gaussian process that captures the distortions is trained. Column 2: Every pixel from the original image is mapped to a new pixel location by the Gaussian process. Every Gaussian process prediction provides both a mean and a variance. In column 2, the variance is depicted. This value is normalised between zero (blue) and one (red).

**Figure 10 sensors-25-00298-f010:**
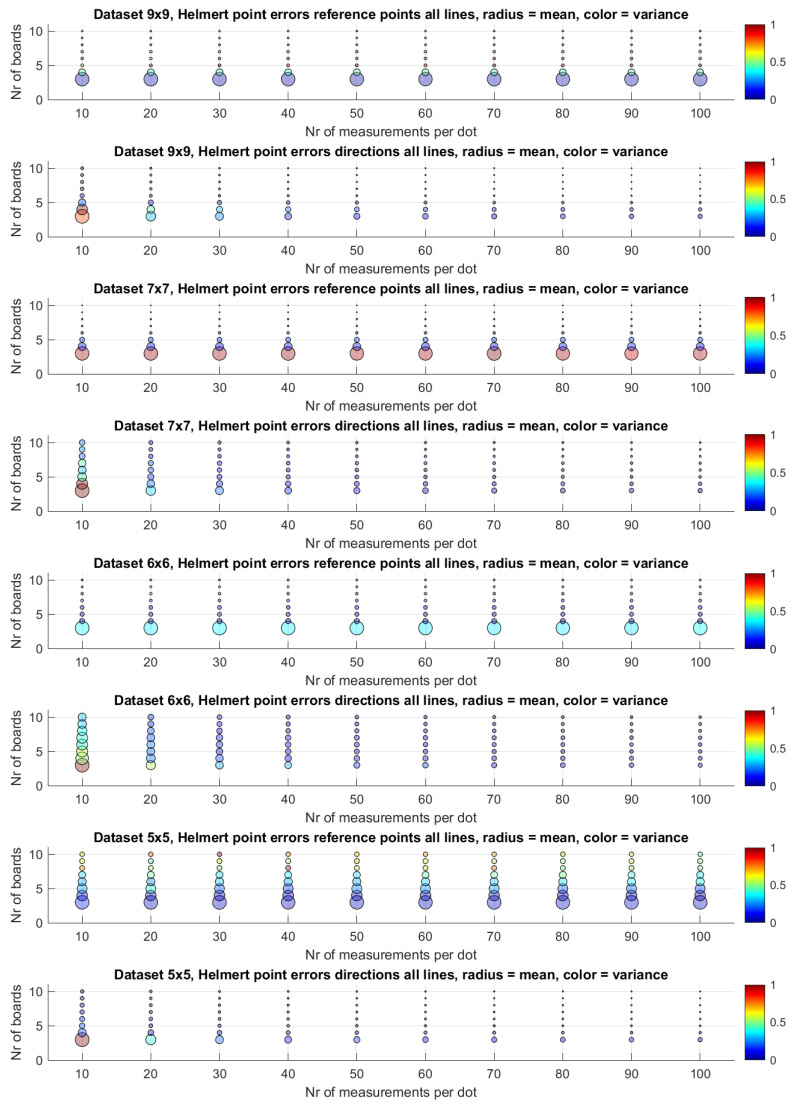
An overview of the Helmert point errors per dataset and per combination. This error can be seen as a measure of residual uncertainty. Every circle depicts an averaged Helmert point error of a resulting set of lines. A larger circle means a larger error. The variance in those errors normalised between 0 and 1 is given by a colour.

**Figure 11 sensors-25-00298-f011:**
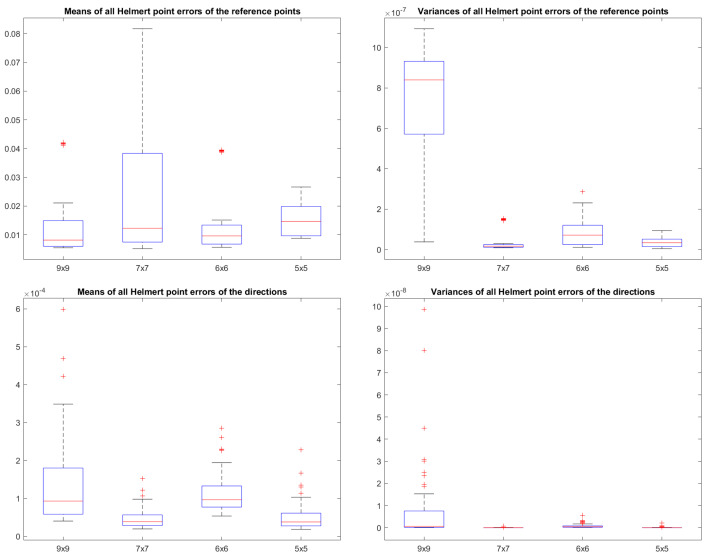
A comparison of the Helmert point errors over all datasets.

**Figure 12 sensors-25-00298-f012:**
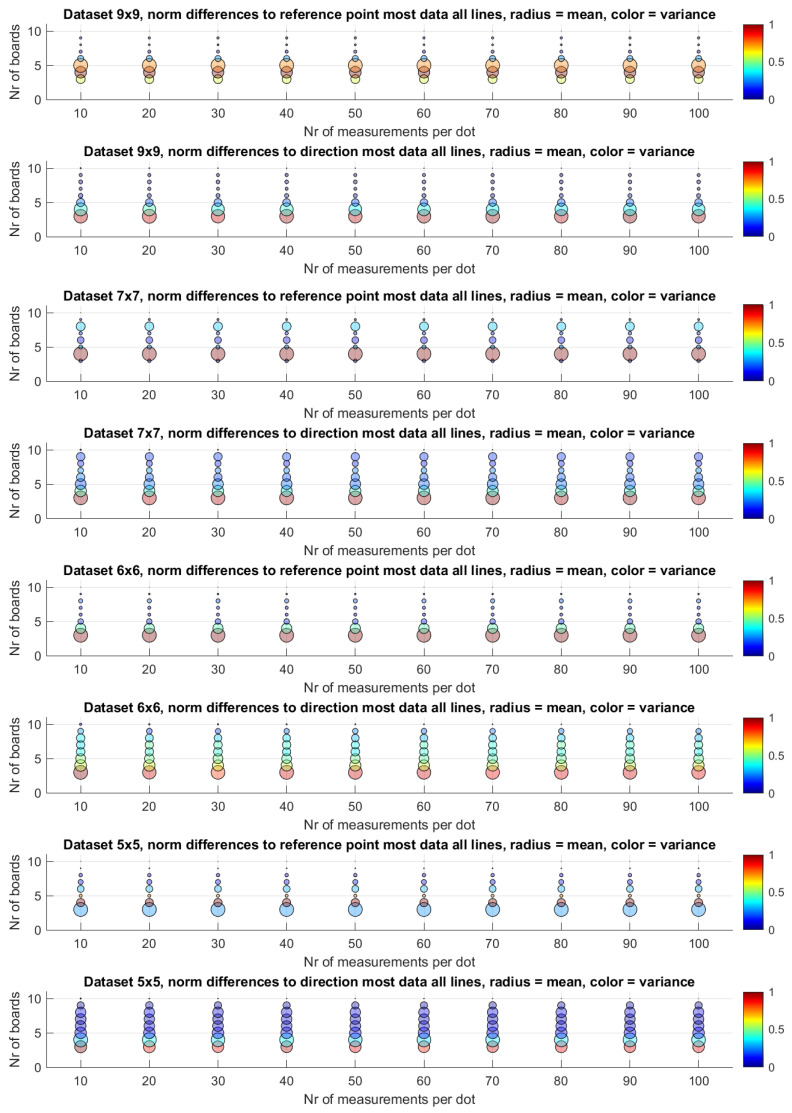
An overview of the differences to the line based on the most available data. This error is calculated as the norm of the difference vector for either the point or the direction, in reference to the lines based on all the data. Every circle depicts an averaged error of a resulting set of lines. A larger circle means a larger error. The variance in those errors, normalised between 0 and 1, is given by a colour.

**Figure 13 sensors-25-00298-f013:**
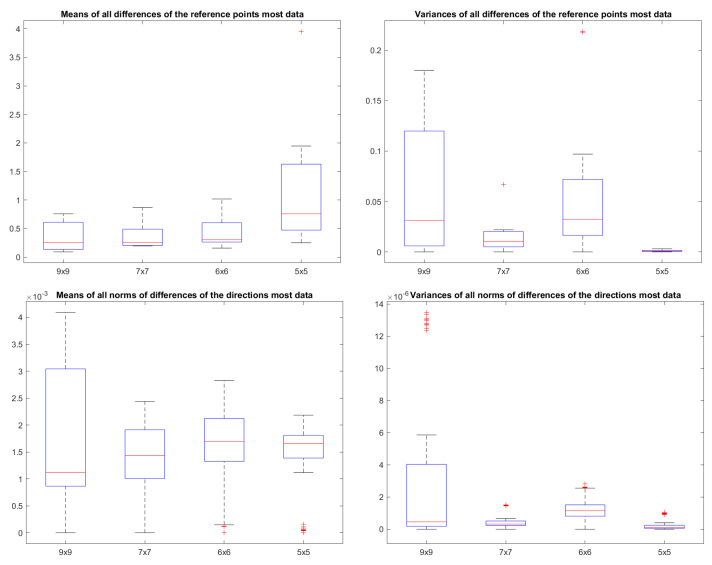
A comparison of the differences of the reference points and directions of all lines with respect to the lines based on the most available data.

**Table 1 sensors-25-00298-t001:** Overview of the details of the datasets. The angles α and β are the rotation angles of the galvanometers in degrees. They result in angles for the directions of the lines that are twice that value, as rotating a mirror by one degree means rotating the reflected laser beam by two degrees.

Name Dataset	Colour Laser	Device	Resolution	Min α	Max α	Min β	Max β	Max GP std
9×9	Red	PSV400	1920×1080	−10	10	−10	10	0.0515
7×7	Green	VibroFlex	2560×1440	1.1	5.5	1.1	5.5	1.4959
6×6	Red	PSV400	2560×1440	2.5	10	−10	10	0.0378
5×5	Red	PSV400	2560×1440	−1	1	−1	1	0.2185

## Data Availability

The data presented in this study are available on request from the corresponding author.

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
