# Peer review of "On the Measurement of Laser Lines in 3D Space with Uncertainty Estimation"

_sensors, 2025, doi:10.3390/s25020298_

Round 1
Reviewer 1 Report
Comments and Suggestions for Authors
The work “On the measurement of laser lines in 3D space with uncertainty estimation” is devoted to the description of a new stochastic method for measuring the coordinates of laser lines in 3D space that can be used on noisy measurements in systems of 3D scanning, laser Doppler vibrometry, LiDAR, etc. The work is complete and well structured. The detailed presentation leaves a very good impression and will be understandable to a wide range of readers. The key part of the work is the Material and Methods section, which describes in detail the processes of camera calibration, measuring 2D and 3D coordinates of points, and constructing lines based on them, taking into account known uncertainties. In the Results section, the proposed method is tested. The authors found that increasing the number of measurements per dot significantly reduces the uncertainty for the line’s direction but has minimal impact on its reference point. The final and intermediate results are interesting. The work was carried out to a high standard and deserves publication in the journal Sensors after correction in accordance with the following comment:
- Despite the detail of the presentation, the work does not compare the results obtained with the results obtained using other methods and approaches known in the literature. This would be useful in order to emphasize the individuality and high qualities of the solution proposed by the authors.
Reviewer 2 Report
Comments and Suggestions for Authors
The overall manuscript is interesting and has special perspective.
However, there are several major problem to be reconsidered as follows:
1. Please explain the GP model in a more specific way, even if the reference [10] has talked something in it. What is the exact difference between classical image undistrotion methods vs. the authors' GP model mentioned in this manuscript? Is the GP model physics based ? Also, in reference [10] the comparison between GP model and MATLAB undistortion algorithm is not that sound.
2. Please compare the proposed method to RANSAC method when determining which extracted dot is valid in the calibration process.
3. What is the exact method to extract the laser dot from the image, how the authors model the noises of laser dot extraction from image.
4. Please compare the proposed method to other SOTA calibration methods in relevant field. Only the results in the current manuscript is not enough to prove the performance of the proposed method.
5. In addition, what is the precision or resolution of the devices that rotates the mirror?
Reviewer 3 Report
Comments and Suggestions for Authors
This manuscript presents a stochastic method for 3D laser line measurement with uncertainty estimation, utilizing minimal hardware and robust statistical modeling to improve calibration accuracy and efficiency. Here are some concerns:
1. Could the authors elaborate on how the method would handle non-Gaussian uncertainties or datasets with significant outliers? Are there plans to adapt the Gaussian process model for such cases?
2. How does the method perform under varying environmental conditions, such as changes in lighting or vibrations? Are there any assumptions that could limit its real-world application?
3. The manuscripts notes diminishing returns after a certain number of boards. How sensitive is the method to smaller datasets, and what is the minimum dataset size for reliable results?
4. What specific advancements in hardware (e.g., higher-resolution cameras) or software (e.g., more sophisticated GP models) could further enhance the accuracy of your method?
5. Include a table summarizing the performance of the proposed method relative to other state-of-the-art techniques across key metrics (e.g., accuracy, cost, time efficiency).
6. The manuscript acknowledges the impact of board placement but does not offer robust guidelines or optimization strategies for practitioners. Provide a detailed subsection offering practical guidelines on the optimal placement of boards and selection of measurement parameters.
7. Add more descriptive captions and annotations to figures, ensuring they can be understood independently of the main text.
Round 2
Reviewer 2 Report
Comments and Suggestions for Authors
There should be some comparison between at least close-field works. Otherwise, the readers can not understand the method proposed in this paper.
